# Physicochemical, Mechanical, and Esthetic Properties of the Composite Resin Manipulated with Glove Powder and Adhesive as a Modeling Liquid

**DOI:** 10.3390/ma15217791

**Published:** 2022-11-04

**Authors:** Jose Bauer, Ruan Pinto Mendes, Rayssa Cavaleiro de Macedo, Edilausson Moreno Carvalho, Leonardo Lopes, Renata Grazziotin-Soares, Darlon Martins Lima, Barbara Costa Oliveira

**Affiliations:** 1Dentistry Biomaterials Laboratory (Biomma), School of Dentistry, Federal University of Maranhão (UFMA), Av. dos Portugueses, 1966, São Luís 65080-805, Brazil; 2Department of Biomaterials and Oral Biology, School of Dentistry, University of São Paulo (FOUSP), Av. Prof. Lineu Prestes, 2227, São Paulo 05508-900, Brazil; 3School of Dentistry, University Ceuma (UNICEUMA), R. Josué Montello, 1, Renascença II, São Luis 65075-120, Brazil; 4Endodontics Division, Department of Oral Biological and Medical Sciences, Faculty of Dentistry, University of British Columbia (UBC), Vancouver, BC V6T 1Z4, Canada

**Keywords:** composite resin, mechanical properties, water sorption and solubility, color stability, thermogravimetric analysis

## Abstract

Composite resins with low flowability are usually handled and manipulated before insertion into the tooth preparation with gloved hands and/or using an instrument covered with a little amount of adhesive to facilitate modeling. We investigated if the modeling techniques (combined or not) affected physicochemical and esthetic properties of a composite resin. Specimens were fabricated and divided into groups according to the handling/modeling technique: Gloved-hands (composite was hand-manipulated with powdered latex gloves); Adhesive (adhesive was used in between the composite layers); Gloved-hands + Adhesive; Control (no adhesive and no touch with gloved-hands). The highest values for flexural strength (MPa), modulus of elasticity (GPa), and fracture toughness (MPa.m^0.5^) were obtained for Adhesive and Gloved-hands + Adhesive (*p* < 0.05); the lowest values were obtained for Control and Gloved-hands (*p* < 0.05). The Control group had the highest sorption. The Gloved-hands (*p* < 0.05) group had the highest solubility. Adhesive and Gloved-hands + Adhesive had a similar solubility (*p* > 0.05). The Control group (*p* < 0.05) had the lowest solubility. There was no statistical interaction between translucency vs. handling/modeling techniques and color stability vs. handling/modeling techniques. Adhesive as a modeling liquid protected the composite against sorption and solubility (if powdered gloves were used) and improved its physical/mechanical properties. Translucency and color stability were not correlated with modeling techniques.

## 1. Introduction

When the dentist performs composite resin restorations, there are several possibilities of contamination of the operative field that can impair the longevity of the dental restoration [1]. The contamination of the composite resin may negatively affect the clinical results for dental patients. The appropriate use of a rubber dam prevents the contamination of the field and material by the patient’s saliva and, eventually, gingival blood. Scientific evidence has shown the increased survival rate of adhesive composite resin restorations when they were performed under rubber dam isolation in relation to the ones that were performed under a cotton roll isolation [2,3]. Nevertheless, due to the high sensitivity of composite resins [4], even using rubber dam isolation, during the manipulation of the material or during its insertion into the dental preparation, there is a chance of contamination and potential negative implications for the restoration.

Depending on the physicochemical characteristics of the composite resin, the dentist may choose different methods of manipulation for specific materials. Composite resins that present low flowability are commonly handled and manipulated with gloved hands before insertion into the preparation [5] or using a spatula (or microbrush) covered with a little amount of adhesive to facilitate modeling [6].

A few studies have suggested that the direct contact of gloved hands on the composite resin will contaminate the material due to the powder added on the latex gloves. The contamination with gloves powder can reduce the cohesive strength [7] and microhardness [8] of the composite, consequently jeopardizing the quality and durability of the restoration [7,9]. Adhesive systems used as lubricants to gain flowability may also impair the restoration, creating small defects and propagating pre-existing cracks in between the incremental layers of the composite resin [10].

Despite the available evidence showing some negative consequences of gloves-powder and adhesives on properties of composite resins, the investigation of a more complete assortment of material properties, and the construction of a more clinical-related in vitro experiment, would supply the strongest evidence to help the selection of modeling techniques.

Therefore, considering that the use of a combination of modeling techniques (gloved-hands and/or adhesive) may facilitate an appropriate adaptation of the composite resin into the tooth preparation, this study investigated if modeling techniques (combined or not) affected the following physicochemical properties of a composite resin: flexural strength/modulus of elasticity, fracture toughness, sorption/solubility, weight loss, translucency, and color stability.

The null hypothesis was that all the tested properties of a composite resin modeled with gloved-hands and/or adhesive would remain identical to the modeling technique using only a spatula (control group).

## 2. Materials and Methods

### 2.1. Materials

The following materials were used: (1) latex gloves containing powder for dental procedures (Supermax Premium Quality, Maxter Glove Manunf. Selangor, Malaysia); (2) hydrophobic portion of an adhesive system (Adper Scotchbond Multi-Purpose conventional three-step adhesive system (3M/ESPE, St. Paul, MN, USA)); (3) composite resin (microhybrid, Filtek Z250 XT, 3M/ESPE, St. Paul, MN, USA). Complete information on the composition of gloves, adhesive, and composite resin is shown in Table 1.

### 2.2. Composite Resin Specimens’ Preparation

The specimens were prepared using three increment-layers of the composite resin inserted into metal matrices with dimensions appropriated for each physicochemical test: strength/modulus of elasticity (25 × 2 × 2 mm), fracture toughness (25 × 5 × 3 mm), sorption/solubility (15 × 1 mm), and translucency/color stability (10 × 1 mm). After accommodation of the 3 increment-layers into the matrix, the specimen was then light-cured in three different areas (in the center and at both ends) for 20 s for each area with a curing light (Radii-Cal curing light, 1200 mW/cm^2^, SDi, Baywaster, Victoria, Australia).

### 2.3. Experimental Groups

The groups of composite resin specimens were divided according to the modeling/handling techniques used to handle/model the material into the stainless-steel matrices, as follows: (1) Gloved/hands group: each increment in composite resin was hand-modeled for 10 s each and inserted into the matrix with a metallic spatula. The operator prepared each specimen wearing a brand-new pair of gloves. (2) Adhesive group: each increment in composite resin was inserted into the matrix with a clean titanium-silicate spatula and modeled using a microbrush (Regular, 2.0 mm, Vigodent, Rio de Janeiro, RJ, Brazil) soaked with adhesive. A brand-new microbrush was used for each specimen preparation. (3) Gloved/hands + Adhesive group: this group is a combination of the previous two groups, i.e., each increment in composite resin was hand-modeled for 10 s each, inserted into the matrix, and modeled using a microbrush soaked with adhesive. (4) Control group: each increment in composite resin was inserted into the matrix with a titanium-silicate spatula (Indusbello Co., Londrina, PR, Brazil). No adhesive and no touch with gloved hands. In between each specimen preparation, the spatula was cleaned with 70% alcohol and dried with absorbent paper.

### 2.4. Flexural Strength and Modulus of Elasticity Analyses

To measure flexural strength and modulus of elasticity of composite resin specimens handled/modeled under different circumstances, *n* = 10 specimens were prepared for each experimental group. Bipartite stainless-steel matrices (ODEME Biotec. Joaçaba, SC, Brazil) with internal dimensions according to ISO 4949 were used to accommodate the composite resin. The matrix was rested on a polyester strip over a 1 mm thick glass slide. The increment-layers of composite resin were accommodated into the matrix. A second set of polyester strips + glass slide were placed over the matrix/specimen. The specimen was light-cured, removed from the matrix, cleaned from the excesses of composite resin (using a scalpel with blade), and immersed in distilled water into a closed recipient (capped and opaque). Specimens remained stored at 37 °C for 24 h. Prior to testing, specimens were measured with a digital caliper (Mitutoyo Corp, Tokyo, Japan) with an accuracy of 0.001.

The three-point bending test was performed on a universal testing machine (Instron 3342 Single Column, Instron, Canton, MA, USA). The distance between the two supports at its ends was 20 mm and the application of the force in the middle was at a speed of 1 mm/min. The flexural strength was calculated using the following formula:$$FS=3xfxl/2xbh^{2}$$
where FS is the flexural strength (MPa), F is the load required for fracture, l is the distance between the supports, and b and h are, respectively, the height and length of the specimen (mm). Data used to obtain the elastic modulus were taken from the straight part of the stress–strain curve in the graph originating from the flexural strength test with digital software (Bluehill, Instron, Canton, MA, USA).

### 2.5. Fracture Toughness Analysis

To measure the fracture toughness of composite resin specimens handled/modeled under different circumstances, *n* = 6 specimens were prepared for each experimental group. Stainless-steel matrices (ODEME Biotec, Joaçaba, SC, Brazil) with internal dimensions according to ASTM E-699 and a slit of 2.8 mm were used to accommodate the composite resin. Identical procedures performed for the previous tests were executed, such as: inserting incremental layers of composite into a set of polyester strips + thin glass slide, light curing, excess removal, storage, and digital caliper measurement. The fracture toughness was calculated using the following formula:K_Ic_ = (PL/bw1,5) ∫ (a/w)
where
∫(a/w) = 3/α(a/w)0.5{1.99 − (a/w)(1 − a/w) × [2.15 − 3.93 a/w + 2.7 (a/w)^2^]}
where
α = 2(1 + 2 a/w)(1 − a/w)3/2
where K_Ic_ = stress intensity factor; P = fracture load; L = distance between the device supports; w = specimen length; b = specimen thickness; a = notch depth.

### 2.6. Sorption and Solubility Analyses

To measure the sorption and solubility of composite resin specimens handled/modeled under different circumstances, *n* = 5 disk-like specimens were prepared for each experimental group according to the ISO 4049:2008. Identical procedures performed for the previous tests were executed: inserting incremental layers of composite into a set of polyester strips + thin glass slide, light curing, and excess removal. The storage comprised immersing the specimens in distilled water at 37 °C for 30 days. Throughout this period, specimens were removed from the distilled water, dried with absorbent paper, and weighted in an analytical scale (AUW 220D, Shimadzu Corp., Kyoto, Japan) at baseline (day-0), day-7, day-14, day-21, and day-30. These measurements provided the sorption values. Specimens were then dehydrated in a vacuum desiccator over 10 days, and subsequently weighted again to obtain the solubility values.

Sorption and solubility were calculated according to ISO 4049 using the formulas:

Sorption = m2–m3/V, where m2 = absorbed mass; mf = final mass; V = volume

Solubility = m1–m3/V, where m1 = initial mass; m3 = final mass; V = volume

The volume was calculated as a mean of three digital caliper measurements (in different points) of area and thickness around the specimen.

### 2.7. Thermal/Weight Loss Analysis

Thermogravimetric analysis (TGA/DSC) for resin composites was performed on a STA-449-C device (Netzsch, Instruments Inc., Burlington, MA, USA) using a platinum pan as a reference material. Analysis was carried out between 38 °C and 600 °C at the heating rate of 10 °C/min under an inert nitrogen environment with a flow rate of 50 mL/min.

### 2.8. Translucency and Color Stability Analyses

To measure translucency and color stability of composite resin specimens handled/modeled under different circumstances, *n* = 6 specimens were prepared for each experimental group. Specimens were prepared using a stainless-steel matrix (calibrated at a 1 mm thickness and a 10 mm central orifice/diameter) (ISO/TR 28642:201151), aiming to create a shade scale (Porcelain Sampler (Smile Line, St-Ilmier, Switzerland)) [11]. A metallic matrix was used to accommodate the incremental layers of composite resin, and the set matrix/composite was gently pressed with a glass slide to guarantee the surface smoothness. Light curing was performed with a Radii Cal (Sdi) curing light for 20 s, according to the manufacturer’s specifications. The distance between the light source and the specimen was standardized to 1.0 mm using a glass coverslip. The tip of the curing light came into contact with the glass slide during the photopolymerization process. Specimens were then stored in small recipients and immersed in water at 37 °C for 6 months. The immersion solution was changed every 10 days.

Color stability of specimens was analyzed by reflectance spectrophotometry (ISO7491:2000), using a spectrophotometer (VITA Easyshade Compact, Vident, Brea, CA, USA) at baseline (day-0), 24-h, day-7, day-90, and day-180.

Translucency of the samples was measured using the translucency parameter method, where the color parameters of each specimen were recorded according to the CIE L* a*b* scale on white and black backgrounds. In this system, L* indicates the luminosity in which the average varies from 0 (black) to 100 (white); a*b* indicates the hue; a* represents saturation in the red-green axis; b* represents saturation in the blue-yellow axis. Translucency was calculated using the following formula:$$TP=\sqrt{{\left(L_{w}^{*}-L_{B}^{*}\right)}^{2}+{\left(a_{W}^{*}-a_{B}^{*}\right)}^{2}+{\left(b_{w}^{*}-b_{B}^{*}\right)}^{2}}$$
where L*_W_, a*_W_, and b*_W_ were measured in the white background, and L*_B_, a*_B_, and b*_B_ were measured in the black background.

Before each measurement, specimens were washed in running water for 10 s and lightly dried with an absorbent paper towel, and the spectrophotometer was calibrated according to the manufacturer’s specifications. Specimens were then positioned on a white and black background and were measured 3 times in each background to obtain an average. Color stability was carried out with the color parameters (∆E) obtained in the translucency test on the white background, which were used to calculate the color change after storage in water, in the same experimental period. ∆E was calculated using the formula:$$\Delta E^{*}=\sqrt{{\left(\Delta L^{*}\right)}^{2}+{\left(\Delta a^{*}\right)}^{2}+{\left(\Delta b^{*}\right)}^{2}}$$
where ∆L*, ∆a*, and ∆b* are the difference between the final and initial color parameters L*, a*, and b*, respectively.

### 2.9. Data Analysis

Statistical analysis was conducted using SigmaPlot 13.0 software (Systat Software Inc., San Jose, CA, USA) All data were subjected to the Shapiro–Wilk test to determine data normality. Data originated from the flexural strength, elastic modulus, fracture toughness, and sorption/solubility tests were compared between groups using One-Way (1-factor) and Holm–Sidak ANOVA for contrast of means (α = 0.05). Data originated from the translucency and color stability tests were analyzed using the two-way repeated-measures ANOVA (manipulation vs. time) and Holm–Sidak tests (α = 0.05). Data originated from the thermogravimetric analysis were descriptively reported: percentage of weight loss in relation to the temperature.

## 3. Results

Data resulting from flexural strength (MPa), modulus of elasticity (Gpa), and fracture toughness (MPa.m^0.5^) tests for composite resin specimens handled/modeled under different techniques are shown in Table 2. For these three experimental tests, the highest values were obtained for Adhesive and Gloved/hands + Adhesive (*p* < 0.05); the lowest values were obtained for Control and Gloved/hands (*p* < 0.05).

Data resulting from sorption and solubility (µg/mm^3^) tests are also shown in Table 2. The highest sorption values were obtained for Control. The other groups (Gloved/hands, Adhesive, and Gloved/hands + Adhesive) had similar sorption values (*p* > 0.05). The highest solubility values were obtained for Gloved/hands, statistically significant (*p* < 0.05) in relation to the other groups; Adhesive and Gloved/hands + Adhesive had similar solubility values (*p* > 0.05); the lowest values were obtained for Control (*p* < 0.05).

The thermogravimetric analysis (TGA) showed a similar total mass loss for the groups: Gloved/hands (21.7%); Adhesive (24.0%); Gloved/hands + Adhesive (25.0%); Control (22.8%). Figure 1 shows the TGA curves for composite resin specimens handled/modeled under different techniques. The weight loss for all samples/groups initiated approximately at 300 °C, and the highest weight loss was observed at 450 °C.

Translucency test results are shown in Table 3. There was no statistical interaction between the main factors (*p* = 0.270). No statistical difference was found for ‘*handling/modeling technique*’ (*p* > 0.05), but a statistical difference was found for ‘*experimental time*’ (*p* < 0.001). Intra-group analysis had the following statistical results for experimental time: baseline/day-0 > 24 h > day-7 > day-90 = day-180 (*p* > 0.583).

Color stability test results are also shown in Table 4 and Table 5. Regarding the ΔL parameter, there was no statistical interaction between the main factors (*p* = 0.355). No statistical difference was found for ‘*handling/modeling technique’* (*p* > 0.05), but a statistical difference was found for ‘*experimental time*’. Intra-group analysis had the following statistical results: 24 h < day-7 < day-90 < day-180. Regarding the ΔE parameter, there was no statistical interaction between the main factors (*p* = 0.065). No statistical difference was found for ‘*handling/modeling technique*’ (*p* > 0.05), but a statistical difference was found for ‘*experimental time*’. Intra-group analysis had the following statistical results: 24 h < day-7 < day-90 < day-180.

## 4. Discussion

This study considered that the use of a combination of modeling techniques (powdered gloved hands and/or adhesive) may facilitate an appropriate adaptation of the composite resin into the tooth preparation. Grounded on this consideration, we investigated if those modeling techniques (combined or not) would affect physicochemical properties of a composite resin.

Our findings showed that the use of a small amount of hydrophobic adhesive in between the incremental layers of the restoration favored the mechanical properties of the composite resin (flexural strength, modulus of elasticity, and fracture toughness). We also found that the use of adhesive protected the composite resin from sorption and solubility. However, gloves powder negatively affected the solubility of the restoration. Thermal analysis found a higher mass loss for composites modeled with adhesive. In addition to that, this study showed no correlation of handling/modeling techniques with translucency and color stability at the 180-days follow-up. Nevertheless, we did reinforce the findings that composite resin loses translucency and increases color stability over time. These findings rejected our null hypothesis as adhesive and gloved hands interfered in some of the tested composite resin properties.

It is noteworthy to say that the adhesive was the contributor to improve physical properties as the handling/modeling technique only with gloved hands had the same results than the control group (no adhesive and no touch with gloved hands). Previous authors have shown that powdered-gloves impaired mechanical properties of resin composites (powdered-gloves were even more damaging than saliva contamination), suggesting then that latex gloves should be cleaned with ethanol when the hand shaping technique is chosen [7]. In our study, powdered gloves were inert, at least in regard to mechanical properties.

Adhesive made the composite more rigid and prone to elastic deformation before breaking or permanently deforming, and more resistant to failure by cracking. These findings are important because in the dental clinic setting, high values of flexural strength, modulus of elasticity, and fracture toughness are expected from posterior composites to withstand the occlusal forces of mastication and preserve the adhesive interface [12]. Reasons for the improved mechanical properties with the use of a small amount of adhesive between the incremental layers of composite might have been the following: (1) adhesive prevented the incorporation of air bubbles in between adhesive-composite layers, (2) adhesive promoted a better interaction between the composite–composite layers, favoring the composite cohesive strength, which, consequently, reduces the air incorporation and defects into a composite layer [5,11,13].

Up until now, studies investigating the sorption, solubility, mass loss, translucency, and color stability of composites modeled with powdered gloved hands and/or adhesives were absent in the dental literature. Our study was the first to consider this interaction and showed that adhesive modeling protected composites from sorption and solubility. One hypothesis that could explain these findings is that the adhesive used in this experiment had a high concentration of hydrophobic monomers, which acted as a protection barrier and reduced the susceptibility of hydrolyzation of the composite resin compounds [14,15,16]. However, even though the adhesive used in this present study contained a high concentration of hydrophobic monomers, some solubility has been reported for these types of adhesives [14], due to their low rates and degrees of conversion: below 70% [17]. This low degree of conversion may be indirectly observed in our thermal analysis, which showed higher degradation for the composite groups modeled with the adhesive system, possibly due to the presence of HEMA [18]. Additionally, our composite samples that were handled/modeled with gloves presented the highest solubility, and this is a consequence of the powder that is added to the latex, which is extremely soluble and prone to water/humidity degradation [19]. Therefore, if using latex gloves (as those were tested in the present study) for manipulation of the composite, powder-free ones should be preferred. Additionally, evaluations of the influence of nitrile gloves on composites are necessary.

Extremely high solubility values for all experimental groups found in the present study can be explained due to methodological variations, such as: specimen size, storage time, storage solution, technical standard used (ISO or ADA), and mainly in the desiccation of the specimen before insertion in storage solution.

Translucency and color change are relevant properties for composites mainly when used in esthetic areas. Ideally, both properties should remain the same over time to benefit our patients. However, in general, composites suffer from degradation because of their polymeric nature, which results in a compromised color appearance over time [6]. In the present study, the results did not show statistical differences between the groups tested; a difference was only found between the evaluation times. This means that powdered gloves and adhesive did not affect esthetic properties of the composite resin. Some of the previous articles on this subject showed that the use of adhesive as modeling liquid for composites reduced the alterations in optical properties caused by staining solutions [6,11,20]. These different results may be originated from the differences in methodology amongst the studies, such as: type of composite resins, fabrication of specimens, type of adhesives, immersion media, and evaluation time. Although our study found that the tested esthetic properties were not influenced by powdered gloves and by a hydrophobic adhesive, we did find that all composite samples lost translucency and increased color stability over time, reinforcing the settled results from the literature [21].

This current study has strength and weaknesses. The main limitation of it is the impossibility to directly transport the findings to the clinical setting, as any other laboratorial experiment [22]. In addition, it may be challenging to create a real clinical scenario using different experimental times that are recommended to evaluate several composite properties (mechanical tests were carried out 24 h after specimens’ preparation; sorption/solubility tests, after 28 days; esthetic tests, after 180 days). Another limitation is the use of only one type of glove and adhesive. It is important to point out that in the clinical setting, the amount of adhesive and the handling techniques may vary between operators, which could lead to different results. Despite those challenges, our results can guide the decision-making process for choosing a modeling technique for composites, in addition to possibly inspiring further in vivo investigations. The practical relevance of our findings is that dentists may use hydrophobic adhesive as a modeling substance (with or without powdered gloves) when they are performing composite restorations under the incremental layer technique and expect no negative consequences for the material.

## 5. Conclusions

In conclusion, the modeling technique using hydrophobic adhesive in between the incremental layers of a composite resin protected the restoration against sorption and solubility (30-days follow-up) and improved the mechanical properties (flexural strength, modulus of elasticity, and fracture toughness) of the composite. The powdered gloved handling/modeling technique did not interfere with the physical/mechanical properties and, also, protected the composite against sorption; however, gloves powder negatively affected the solubility of the restoration. Translucency and color stability (180-day follow-up) were not correlated with gloved-hands and adhesive modeling techniques.

In sum, our findings, even in light of the limitations of an in vitro study, suggest that modeling techniques using adhesives favor physicochemical properties of composite resins. Finally, adhesives, used in between the incremental layers of the restoration, are not expected to interfere with esthetic properties (translucency and color stability) of the material. Touching composites with powdered latex gloves makes the composite more soluble.

## Figures and Tables

**Figure 1 materials-15-07791-f001:**
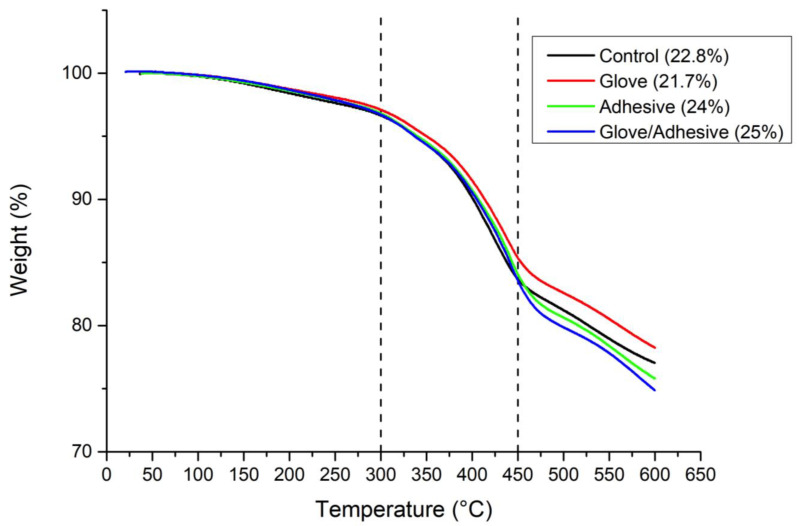
Thermogravimetric analysis (TGA) of composite resin specimens handled/modeled under different techniques (gloved-hands and/or adhesive): weight loss x temperature.

**Table 1 materials-15-07791-t001:** Complete information on the composition of gloves, adhesive, and composite resin used in this study *.

Material	Manufacturer	Composition
Filtek Z250 XT Composite Resin	3M/ESPE, St. Paul, USA	Silane treated ceramic (70–85%), Bisphenol A diglycidyl ether dimethacrylate (BisGMA 1–10%), Bisphenol A polyethylene glycol diether dimethacrylate, Diurethane dimethacrylate (UDMA 1–10%), silane-treated silica (1–10%), Triethylene glycol dimethacrylate (TEGDMA <1%)
Adper Scotchbond Multi-Purpose Adhesive	3M/ESPE, St. Paul, Mn, USA	Bisphenol A diglycidyl ether dimethacrylate (BisGMA 55–65%), 2-Hydroxyethyl Methacrylate (HEMA 35–45%)
Supermax Gloves	Maxter Glove Manunf. Selangor, Malaysia	Natural Rubber Latex, Zinc Oxide, Sulfur, Blocked Phenol, Titanium Dioxide, zinc diethyldithiocarbamate, Micro-Refined Wax Emulsion, Potassium Hydroxide, Calcium Carbonate, Calcium Nitrate, Nitric Acid, Corn starch.

* Information collected from the manufacturers’ websites and package’s instructions.

**Table 2 materials-15-07791-t002:** Data (mean ± standard deviation) resulting from flexural strength (MPa), modulus of elasticity (Gpa), and fracture toughness (MPa.m^0.5^) tests for composite resin specimens handled/modeled under different techniques (gloved-hands and/or adhesive) *.

Groups	Modulus of Elasticity (Gpa)	Flexural Strength (MPa)	Fracture Toughness (MPa.m^0.5^)	Sorption (µg/mm^3^)	Solubility (µg/mm^3^)
Control	9.3 ± 0.8 b	140.5 ± 12.3 b	1.19 ± 0.04 b	22.9 ± 1.6 a	8.7 ± 2.3 c
Gloves	10.4 ± 1.0 b	130.2 ± 14.19 b	1.21 ± 0.08 b	14.3 ± 0.5 b	15.7 ± 0.5 a
Adhesive	13.2 ± 2.0 a	165.9 ± 25.5 a	1.38 ± 0.05 a	16.4 ± 0.3 b	11.6 ± 0.3 b
Gloves/Adhesive	15.0 ± 2.0 a	171.3 ± 23.7 a	1.46 ± 0.08 a	15.3 ± 0.8 b	13.1 ± 1.1 b

* Similar lowercase letters in the same column indicate statistical difference (*p* < 0.05).

**Table 3 materials-15-07791-t003:** Data resulting from translucency test for composite resin specimens handled/modeled under different techniques (gloved-hands and/or adhesive), in different evaluation times *.

	Translucency				
	Gloves	Adhesive	Gloves Adhesive	Control	
Baseline	9.2 ± 1.1	10.5 ± 1.1	10.5 ± 0.8	9.9 ± 0.5	a
24 h	6.6 ± 1.8	8.3 ± 1.3	6.8 ± 0.8	7.1 ± 1.7	b
Day-7	4.8 ± 1.1	6.8 ± 1.4	5.0 ± 1.2	6.3 ± 2.5	c
Day-90	1.7 ± 1.0	4.0 ± 1.6	1.8 ± 0.5	2.6 ± 1.2	d
Day-180	3.2 ±1.6	2.9 ± 2.0	2.3 ± 0.8	2.6 ± 2.1	d

* Similar lowercase letters indicate statistical difference between the different evaluation times for each tested property (*p* < 0.05).

**Table 4 materials-15-07791-t004:** Data resulting from color stability test **(ΔE)** for composite resin specimens handled/modeled under different techniques (gloved-hands and/or adhesive), in different evaluation times *.

	Color Stability (ΔE)				
	Gloves	Adhesive	Gloves Adhesive	Control	
Baseline	14.7 ± 3.6	16.5 ± 10.0	8.4 ± 1.6	11.1 ± 2.0	d
24 h	30.6 ± 2.3	24.0 ± 9.9	16.1 ± 2.9	24.4 ± 2.3	c
Day-7	51.1 ± 8.3	47.3 ± 6.3	41.0 ± 7.0	47.3 ± 2.2	b
Day-90	51.0 ± 7.5	54.5 ± 4.8	49.8 ± 7.3	54.2 ± 3.2	a
Day-180	14.7 ± 3.6	16.5 ± 10.0	8.4 ± 1.6	11.1 ± 2.0	d

* Similar lowercase letters indicate statistical difference between the different evaluation times for each tested property (*p* < 0.05).

**Table 5 materials-15-07791-t005:** Data resulting from color stability test **(ΔL)** for composite resin specimens handled/modeled under different techniques (gloved-hands and/or adhesive), in different evaluation times *.

	Color Stability (ΔL)				
	Gloves	Adhesive	Gloves Adhesive	Control	
Baseline	−13.0 ± 3.5	−8.2 ± 2.6	−7.2 ± 1.6	−7.8 ± 2.1	d
24 h	−21.7 ± 3.5	−12.5 ± 3.2	−11.7 ± 4.0	−15.6 ± 2.7	c
Day-7	−42.6 ± 7.3	−36.1 ± 6.7	−36.1 ± 7.5	−36.2 ± 4.3	b
Day-90	−45.0 ± 6.3	−45.7 ± 4.1	−44.7 ± 8.2	−46.5 ± 5.0	a
Day-180	−13.0 ± 3.5	−8.2 ± 2.6	−7.2 ± 1.6	−7.8 ± 2.1	d

* Similar lowercase letters indicate statistical difference between the different evaluation times for each tested property (*p* < 0.05).

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
