# Peer review of "Physicochemical, Mechanical, and Esthetic Properties of the Composite Resin Manipulated with Glove Powder and Adhesive as a Modeling Liquid"

_materials, 2022, doi:10.3390/ma15217791_

Round 1
Reviewer 1 Report
Dear authors,
congratulations for the interesting topic approached - the handling of the composite material in the direct restoration technique is essential for the success of the restoration. The approached topic is interesting and not very much analyzed in the literature, and the way of handling composite materials must be studied from as many points of view.
I consider that in order to detail some aspects, the authors must specify:
- how was the number of samples calculated for each test so that the results were statistically significant?
- consider that a recommendation can be made regarding the handling of composite materials - to be made only with latex gloves?
- the use of adhesive as a modeling aid can be considered for any type of composite material with low flowability?
The conclusions were formulated based on the results obtained.
The references are in agreement with the chosen subject.
Reviewer 2 Report
The manuscript titled „Physicochemical, mechanical and esthetic properties of the composite resin manipulated with glove powder and adhesive as a modeling liquid” aimed to investigate the influence of handling technique of packable resin composite on its physicochemical and esthetic properties.
The study seems clinically relevant, since various composite resin modelling techniques and the equipment used may influence the materials’ properties. However, manual modelling of resin composite should be limited in the clinical setting.
The authors performed great deal of tests to check the materials properties.
Authors concluded that using adhesive as a modeling liquid protected the composite against sorption and solubility (30-days follow up) and improved its physical/mechanical properties (flexural strength, modulus of elasticity and fracture toughness).
In the Abstract, it was stated that there was no significant differences between study groups (p>0.05), hence the conclusion seems not fully justified. Please rephrase the Abstract and the way in which you described the results in section 3, as it is misleading.
Also, the statement about solubility is not clear, as the solubility in all experimental groups significantly dropped in comparison to the control group.
Please correct the abstract as it falsely presents the results of the study.
Please address the following issues:
Introduction
Please read the following paper https://www.mdpi.com/1996-1944/15/11/3759 and enhance your paper in both, Introduction and Discussion sections.
Methods
What amount of adhesive was used for each increment of resin composite?
Results
Table 3 is difficult to read
Discussion
Some statements are not fully justified in the light of the findings of the present study and should be corrected, e.g.:
“Therefore, if the clinician chooses to hand-manipulate the composite to homogenize it and facilitate its insertion and accommodation into the preparation, powder-free nitrite gloves should be preferred”
The study used only one type of gloves so the recommendation as to which gloves to use is not grounded. Please rephrase the statement and add appropriate references.
Discuss your findings, e.g.:
- that using powdered latex gloves did not negatively influence the physical properties of modelled resin composite.
- water solubility was higher for all experimental groups and contrast it with the literature
The limitations of the study must be revised; e.g. the study used only one type of gloves, only one type of adhesive, etc.
The statement:
“The practical relevance of our findings is that dentists may use adhesive as a modeling substance when they are performing composite restorations under the incremental layer technique and expect no negative consequences for the material.”
is not fully justified and should be more specific as there are different adhesives available on the market and not all of them act the same way as the tested one. Please refer to the paper https://www.mdpi.com/1996-1944/15/11/3759
Minor issues
Please correct grammar and style
e.g. what is meant by “digital modeling technique”?
Check typos, e.g. “powder-free nitrite gloves”; also typos in brand names of the tested products e.g. Adper Scotchbond; Check again Figure 1 description
check the spacings
Referencing style should be adjusted to the Journal’s guidelines.
Reviewer 3 Report
Dear Authors,
The manuscript clearly depicts the hard work involved, however, the attached comments need to be addressed.

Round 2
Reviewer 2 Report
Thank you for revising the manuscript.
Some issues need to be more stressed:
1. As for the corrected statement:
“Therefore, if the clinician chooses to manually manipulate the composite to homogenize it and facilitate its insertion and accommodation in the preparation, powder-free gloves should be preferred. However, evaluations of the influence of nitrile glove on composites are necessary.”
it should be underlined that if using latex gloves (as those were tested in the present study) for manipulation of the composite, powder-free ones should be preferred.
2. Using powdered latex gloves did not negatively influence the physical properties of modelled resin composite in comparison to the control; please contrast it with the literature in Discussion
3. Water solubility of tested composite was higher for all experimental groups in comparison to the control; please contrast it with the literature in Discussion
4. Limitations of the study could be more broadly described.
